# Sleep-Disturbance-Induced Microglial Activation Involves CRH-Mediated Galectin 3 and Autophagy Dysregulation

**DOI:** 10.3390/cells12010160

**Published:** 2022-12-30

**Authors:** Liyang Guo, Kirstin M. Reed, Ashley Carter, Yan Cheng, Soheil Kazemi Roodsari, Damian Martinez Pineda, Laurie L. Wellman, Larry D. Sanford, Ming-Lei Guo

**Affiliations:** 1Drug Addiction Laboratory, Department of Pathology and Anatomy, Eastern Virginia Medical School, Norfolk, VA 23507, USA; 2Center for Integrative Neuroscience and Inflammatory Diseases, Eastern Virginia Medical School, Norfolk, VA 23507, USA; 3Sleep Research Laboratory, Department of Pathology and Anatomy, Eastern Virginia Medical School, Norfolk, VA 23507, USA

**Keywords:** sleep fragmentation, microglia, neuroinflammation, autophagy, CRH

## Abstract

Chronic sleep disturbances (CSDs) including insomnia, insufficient sleep time, and poor sleep quality are major public health concerns around the world, especially in developed countries. CSDs are major health risk factors linked to multiple neurodegenerative and neuropsychological diseases. It has been suggested that CSDs could activate microglia (Mg) leading to increased neuroinflammation levels, which ultimately lead to neuronal dysfunction. However, the detailed mechanisms underlying CSD-mediated microglial activation remain mostly unexplored. In this study, we used mice with three-weeks of sleep fragmentation (SF) to explore the underlying pathways responsible for Mg activation. Our results revealed that SF activates Mg in the hippocampus (HP) but not in the striatum and prefrontal cortex (PFc). SF increased the levels of corticotropin-releasing hormone (CRH) in the HP. In vitro mechanism studies revealed that CRH activation of Mg involves galectin 3 (Gal3) upregulation and autophagy dysregulation. CRH could disrupt lysosome membrane integrity resulting in lysosomal cathepsins leakage. CRHR2 blockage mitigated CRH-mediated effects on microglia in vitro. SF mice also show increased Gal3 levels and autophagy dysregulation in the HP compared to controls. Taken together, our results show that SF-mediated hippocampal Mg activation involves CRH mediated galectin 3 and autophagy dysregulation. These findings suggest that targeting the hippocampal CRH system might be a novel therapeutic approach to ameliorate CSD-mediated neuroinflammation and neurodegenerative diseases.

## 1. Introduction

Sufficient good quality sleep is crucial for maintaining human physical and psychological health. However, in modern society, chronic sleep disturbances (CSDs) including insomnia, insufficient sleep time, and poor sleep quality are highly prevalent due to social factors including competition, sustained stress, and economic pressure, etc. [1] as well as a variety of health problems [2,3]. It has been estimated that nearly 25% of adults in USA have sleep problems [4]. CSDs have also been identified as major risk factors contributing to neurological impairments such as memory and cognitive loss evident in neurodegenerative diseases [5,6] and accelerating the pathogenesis of neuropsychiatric diseases such as anxiety and depression [7,8]. The underlying mechanisms responsible for how CSDs contribute to brain dysfunctions are complex and remain mostly unknown.

Microglia, the brain immunocompetent cells, constitute the first line of defense in response to a variety of external or internal insults including viral invasion, abused drugs, and brain injuries [9,10,11]. Upon stimulation, microglia quickly increase the production of pro-inflammatory mediators and cytokines such as il1β, tnfα, il6, and ccl2 and secret them into the surrounding microenvironment where they interact with neurons and astrocytes. Transient and controlled inflammation is beneficial for neuron and brain repair while uncontrolled and prolonged neuroinflammation can exacerbate existing neuronal and brain damage. Indeed, abnormal microglial activation has been implicated in neurogenerative diseases such as Parkinson’s disease, Alzheimer’s disease, and multiple sclerosis [12,13,14] whereas microglial inhibition has been shown to be a promising therapeutic approach for mitigating these diseases [15,16].

Accumulating evidence demonstrates that CSDs can increase neuroinflammation. In multiple rodent models, microglia were activated by various sleep deprivation regimens and abnormal microglial activation was implicated in CSD-mediated synaptic loss and neuronal dysfunction [17,18]. For example, sleep deprivation for 48 h in rats increased IL1β, TNFα, IL6 levels and decreased the anti-inflammatory factors, IL4 and IL10, in the hippocampus (HP), and led to spatial memory impairment [19] whereas microglia inhibition mitigated spatial memory loss [20]. Sleep restriction (20 h for 10 consecutive days) increased microglial Iba1 levels in the HP in C57BL/6 mice [21]. Chronically sleep-restricted mice and rats show similar changes in neuroimmune signaling and neuronal dysfunction [22,23]. Interestingly, administration of minocycline, an inhibitor of microglial activation, blocked some effects of sleep deprivation [24].

Several neuroimmune pathways have been implicated in CSD-mediated microglial activation. Increased levels of reactive oxygen species (ROS), well-known inducers of microglial activation [25], were reported to be responsible for learning and memory impairment in sleep deprived rats [26,27]. Enhanced activity of the NOD-, LRR- and pyrin domain-containing protein 3 (NLRP3) inflammasome has been identified in CSD-mediated microglial activation and NLRP3 inhibition has beneficial effects on rats with CSDs [28]. However, the endogenous and upstream signal(s) directly activating microglia in the context of CSDs remain unidentified.

Corticotrophin-releasing hormone (CRH) is a neuronal hormone regulated by a variety stress types including sleep disturbances [29,30]. CRH and its cognate receptors 1 and 2 (CRHR1/2) are expressed in both hypothalamus and extra-hypothalamic regions including the striatum, HP, cortex, and amygdala [31,32,33]. Extra-hypothalamic CRH, especially in the HP, is also sensitive to stress and sleep disturbances [34]. Chronic combined stressors increase CRH and neuroinflammation in the HP [35]. Long-term exposure to noise for 30 consecutive days significantly increased the expression of CRH and CRHR1 levels in the HP [36]. One week of running wheel (12 h/day) increased CRHR1 mRNA levels in the HP which is related to depression- and anxiety-related behaviors [37]. Interestingly, CRH has been shown to activate microglia in vitro [38,39] but the underlying mechanisms remain unknown.

To unravel the mechanisms underlying microglial activation induced by CSDs, we exposed mice to three weeks of sleep fragmentation (SF), a common sleep problem [40,41,42], and investigated the status of microglial activation, galectin 3 (Gal3) level, and autophagy in various brain regions including the HP, striatum, and prefrontal cortex (PFc). Our results demonstrated that SF could activate microglia, increase CRH levels, and dysregulate autophagy processes specifically in the HP. In vitro, CRH activates microglia involves Gal3 upregulation and autophagy dysregulation. Taken together, our data revealed a novel mechanism underlying CSD-mediated microglial activation which suggests that targeting CRH-mediated signaling could mitigate CSD-mediated neuroinflammation and related neurological symptoms.

## 2. Materials and Methods

### 2.1. Animal and Reagents

We purchased wild type C57BL/6 mice (male, 10-week-old, 25–30 g on arrival) from Charles River Laboratories (Wilmington, MA). The mice were housed and kept in a colony room with food and water available ad libitum. The colony room was maintained on a 12:12 light to dark cycle and ambient temperature at 23.0 °C ± 1.5 °C. All procedures were conducted in accordance with the National Institutes of Health’s Guide for the Care and Use of Experimental Animals and were approved by Eastern Virginia Medical School’s Institutional Animal Care and Use Committee (protocol number: 20-010). The CRHR2 specific antagonist, As-2B, was purchased from Tocris (catalog# 2391).

### 2.2. SF Procedure

We performed SF by using commercial, validated devices (Lafayette Instruments, Sleep Fragmentation Chamber, model 80391) that employ an automated sweeper arm moving across animal cages. In brief, the mice were randomly divided into two groups receiving sham/SF (n = 18 ± SF). All mice were placed into the devices one day prior to the start of SF to let them become familiar with the environment of device. For SF, sleep was interrupted at 2 min intervals during the 12 h light period (7 AM–7 PM). This SF protocol in mice has been shown to produce moderate to severe SF [40,41,42] without significantly reducing overall sleep or impacting sleep macro- or micro-architecture [41,42]. During the 12 h dark period, the motorized sweeper was stopped letting the mice be free to behave normally. The mice were observed daily to assess their health. Sham animals were maintained in their home cages without any interruption. The mice were subjected to three-weeks of SF and sacrificed one day after the last SF session. The mice were randomly assigned into three subgroups (sub1-3, n = 6): the brains of sub1 mice were subjected to protein/RNA extractions, the brains of sub2 mice were used for slide preparation, and the brains of sub3 mice were used for adult Mg isolation (in striatum, HP, and PFc). Liver, spleen, and gut were collected for total RNA extractions (n = 6).

### 2.3. Adult Microglia Isolation

Mice were anesthetized with 4% isoflurane and transcardially perfused with 1X PBS followed by brain removal. The brains were then y dissected into different regions including the striatum, HP, and PFc. The brain parts were pooled for adult microglia isolation (MACS dissociation kits, Miltenyi Biotech Company, Bergisch Gladbach, Germany). Briefly, around 300–500 mg of brain tissue were homogenized in 2 mL enzyme mixture by using a gentleMACS™ Octo Dissociator (Miltenyi Biotech Company, Bergisch Gladbach, Germany) at 37 °C for 30 min. The homogenates were then transferred to MACS^®^ Smart Strainer with centrifuging at 300× *g* for 10 min at 4 °C. The pellets were then processed for debris and red blood cell removal and dissolved in 500 µL labelling solution. Subsequently, the acquired cells were incubated with 15 µL CD11b beads for 15 min at 4 °C. The labeled cells were sent through a column in a magnetic field. The purified microglia were then suspended in 1 mL 1X PBS with 0.5% BSA and quantified by Countess 3 (Thermo Fisher, Waltham, MA, USA). The CD11b positive cells were seeded into 24-well plates for Ib1 immunostaining. The remaining purified microglia were stored in a freezer at −80 °C for later biochemical analysis.

### 2.4. Primary Microglia and BV2 Cell Culture

Primary microglia (PM) were obtained from 1- to 3-d-old C57BL/6 newborn pups. The brains were removed and cut into small pieces in Hank’s buffered salt solution (Invitrogen, 14025076). Then, the brain tissues were incubated in 0.25% trypsin (Invitrogen, 25300-054) for 15 min. After filtration with 70 µM strain, the mixed glial cultures were resuspended in DMEM (Invitrogen, 11995-065) with 10% heat-inactivated FBS (Invitrogen, 16000-044) with 100 U/mL penicillin, and 0.1 mg/mL streptomycin. We placed PM at 20 × 10^6^ cells/flask density onto 75 cm^2^ cell culture flasks. The cell medium was replaced every 3 d, and after the first medium change, macrophage colony-stimulating factor (Invitrogen, PHC9504) 0.25 ng/mL was added. When confluent (7 to 10 d), mixed glial cultures were shaken gently by hand to promote microglia detachment from the bottom. We then collected the floating PM from each flask and centrifuged them at 1000× *g* for 5 min. PM were plated on 6-well plates for all subsequent experiments. The purity of microglial cultures was evaluated by Iba 1 immunostaining (Wako Pure Chemical Industries, 019-19741).

The BV2 cell line was obtained from Dr. Sanjay Maggirwar (University of Rochester Medical Center, Rochester, NY, USA) and was grown and routinely maintained in DMEM with 10% FBS at 37 °C and 5% CO_2_ and used up to passage 30.

### 2.5. Microglia Treatment

PM and BV2 cells were cultured in vitro with DMEM (10% FBS). Microglia were seeded into 6-well plates with 80% confluence and cultured overnight. The next day, microglia were replaced with fresh DMEM (FBS free) and exposed to CRH with varying doses (25–200 nM) for 6 or 24 h. For pre-treatment, microglial cells were exposed to As-2B at 100 nM for 45 min followed with CRH exposure. At indicated time points, microglial cells were collected either for RNA or protein extractions.

### 2.6. RNA Extraction, Reverse Transcription, and Quantitative (Q) Polymerase Chain Reaction

Around 100–200 mg of brain tissue or 1 × 10^6^ microglia were directly added to 1 mL Trizol (Invitrogen). Brain or microglial lysates were briefly sonicated (3–5 s) with 70% power amplitude on ice and incubated for 20 min on ice. The lysates were centrifuged at 10,000× *g* for 10 min and the supernatants were transferred into new 1.5 mL microcentrifuge tubes with 0.2 mL of chloroform added. After vigorous vertexing, the lysates were centrifuged at 10,000× *g* for 15 min at 4 °C. The upper aqueous phase was transferred to a new tube following with 500 µL isopropyl alcohol add. Samples were incubated at room temperature for 10 min and centrifuged at 10,000× *g* for 10 min to precipitate total RNA. The RNA was washed with 75% ethanol. After 15 min drying at room temperature, the total RNA was dissolved in DEPC-treated H_2_O and quantified by Nanodrop. Reverse transcription (RT) reactions were performed using a Verso cDNA kit (Invitrogen). The reaction system (20 µL) contained 4 µL 5X cDNA synthesis buffer, 2 µL dNTP mix, 1 µL RNA primer, 1 µL RT enhancer, 1 µL Verso enzyme Mix (Invitrogen), total RNA template 1500 ng, and a variable volume of water. Reaction conditions were set at 42 °C for 30 min following with 85 °C for 5 min. QPCRs were performed using One-Step advanced qRT-PCR Kits (Invitrogen). Reaction systems were set up as follows: 10 µL Master mix, 1.0 µL primers, and 1 µL cDNA and 8 µL distilled H_2_O. A QS3 qPCR machine (Invitrogen) was employed for qRT-PCR Mouse primers for tnfα il6, il1β, ccl2, and CRH were purchased from Invitrogen (Mm00443258, Mm00446190, Mm00434200, Mm00441242, and Mm01288386, respectively). Mouse GADPH (Invitrogen, Mm99999915) was used as internal control for RNA quantification.

### 2.7. Western Blots

Around 50–100 mg brain tissue or one 6-well plate microglia were dissolved in 200 µL RIPA buffers with proteinase and phosphatase inhibitors (Thermo Fisher Scientific, Waltham, USA) followed with sonication for 5 s on ice with 70% amplitude (Thermo Scientific). The brain or cell homogenates were incubated at 4 °C for 30 min and then centrifuged at 12,000× *g* for 10 min. The protein supernatants were taken out and quantified through the BCA method. Equal amounts of the proteins (40 µg) were electrophoresed in a sodium dodecyl sulfate-polyacrylamide gel for one hour at 160 V and transferred to immobile PVDF membranes at 180 mA for 90 min. The blots were blocked with 3% nonfat dry milk in Phosphor-buffered saline (PBST). The Western blots were then incubated with indicated antibodies at 3% non-fat milk overnight at 4 °C. The next day, the PVDF membranes were washed three times with PBST and incubated with IRDye fluorescent mouse or rabbit second antibody for 60 min at room temperature. After three washes with PBST, the membranes were put into the Odyssey^®^ Imaging System (Li-Cor Company, USA) for image development and the intensity of target fluorescent band was quantified using Image Studio™ Software (Li-Cor Company, Lincoln, USA). After imaging, the membranes were washed and blocked and re-probed with β-actin for normalization. The following antibodies were used: microglial marker CD11b (1:2000, NBP2-19019), astrocyte marker GFAP (1:5000; Abcam, ab7260), NLRP3 (1:2000, adipogen, AG-20B-0014-C100), Iba1 (1:2000, NBP2-19019), beclin1 (1:2000, NB500-249), LC3B (1:2000; NB100-2220), SQSTM1 (1:2000; Novus Biological, H00008878-M01); LAMP2 (1:2000; NB300-591); TFEB (1:2000; cell signaling technology, #4240), LAMP1 (1:2000; NB120-19294); CRH (1:2000, Abcam, ab184238); CRFR1 (1:1000; Sigma, SAB4500465); CRFR2 (1:1000, Sigma, SAB4500467); galectin 3 (1:2000, Abcam, AB2785), cathepsin B (Abcam, AB214428). Second antibodies were bought from Li-COR including IRDye^®^ 680RD Donkey anti-Mouse (1:5000) or rabbit IgG; IRDye^®^ 800CW Donkey anti-Mouse or rabbit IgG (1:5000). β-actin was purchased from Santa Cruz (1:2000, sc-8432) or from Sigma (1:2000, A2066).

### 2.8. Immunofluorescence Staining

SF/Sham experienced mice were anesthetized with 4% isoflurane and transcardially perfused with 1X PBS followed fixed with 4% PFA. The brains were taken out and put into 4% PFA solution overnight at 4 °C. The brains were then washed with 1X PBS three times. The brains were embedded and sectioned (5 µM) by the EVMS histology core facility. Brain sections were co-incubated with primary anti-Iba1 antibody (1:500, Wako Pure Chemical Industries, Osaka, Japan, 019-19741) or anti-GFAP antibody (1:500, ab7260, abcam) overnight at 4 °C. Secondary AlexaFluor 488 goat anti-rabbit IgG (A-11008) or AlexaFluor 594 goat anti-mouse (A-11032) (Thermo Fisher Scientific Waltham, MA, USA) was added for two hours to detect Iba1 and GFAP, followed by mounting of sections with prolong gold antifade reagent with 4,6-diamidino-2-phenylindole (Thermo Fisher Scientific, Waltham, MA, USA, P36935). Fluorescent images were acquired on a Zeiss fluorescent microscopy. ZEN pro software (Carl Zeiss, Thornwood, NY, USA) was employed to process and analyze the intensity of Iba1 and GFAP signals. For the fluorescence intensity quantification of Iba1 or GFAP, two slices picked from per mouse were imaged and quantified for three mice in each group. Images were obtained under identical exposure conditions (20 X magnitude). All Iba1+ or GFAP+ cells were detected based on the threshold fluorescence intensity of each cell, soma diameter and manual counting tool within the counting frame of the interested regions. Then, by automated measurement, the mean intensity of each soma was measured and data were compared between sham and SF groups and shown in fold changes.

### 2.9. Immunocytochemistry

PM or BV2 cells were plated on 24-well plates with coverslips. The next day cells were fixed with 4% PFA for 15 min at room temperature. Then, cells were treated with 1X PBS with 0.3% Triton X-100 (Fisher scientific, BP151-1). Cells were then incubated with a blocking buffer containing 10% normal goat serum in PBS for 1 h at room temperature followed by addition of rabbit anti-LC3B (1:300) or galectin 3 (1:500) or cathepsin B (1:250) antibody and incubated overnight at 4 °C. The next day, secondary Alexa Fluor 488 goat anti-rabbit IgG or anti-mouse IgG (Invitrogen, A11008) were added at a 1:500 dilution for two hours at room temperature. Cells were then washed 3 times in buffer and mounted with prolong gold antifade reagent with 4,6-diamidino-2-phenylindole (Invitrogen, 36935). Fluorescent images were acquired on a Zeiss fluorescent microscopy under the same exposure conditions and analyzed by ZEN lite software. For Gal3, LC3B, and Cat B puncta analysis, the average numbers of puncta per cells were calculated by the total puncta numbers divided by the total cell numbers in each field. The puncta numbers were counted manually in six randomly selected fields for each sample and at least 50 cells were selected for each treatment.

### 2.10. Statistical Analysis

All data are expressed as means ± the standard error of the mean (SEM). Data were statistically evaluated by unpaired student-t tests or one-way ANOVAs using GraphPad Prism 9 (La Jolla, CA, USA). Tests with probability levels of <0.05 were considered statistically significant.

## 3. Results

### 3.1. Three-Weeks of SF Activates Microglia in the HP and Increases Peripheral Inflammation Levels

Previous studies revealed that chronic SF (two months) increases neuroinflammation in HP and PFc, implying Mg activation [43]. In this study, we aimed to explore the effects of three-weeks of SF directly on Mg in different brain regions. WT mice with three weeks of SF (12 h/per day) were employed for isolating Mg from various brain regions including the striatum, HP, and PFc and the expression of four pro-inflammatory factors (il1β, il6, tnfα, and ccl2) were monitored in purified adult Mg. We also collected liver, spleen, and gut to investigate the effects of SF on peripheral-inflammation levels. Our research schematic is illustrated in Figure 1A and the purity of microglia was confirmed by Iba1 staining (Figure 1B). Significant enrichment of these four pro-inflammatory mediators in purified adult Mg (several hundred to several thousand folds) was observed (n = 6, * *p* < 0.05; Figure 1C,D) confirming that Mg are the main resource for neuroinflammation in vivo. We then investigated the activation status of purified microglia with/without SF. In the striatum, SF-experienced Mg increased *il6* levels (n = 6, * *p* < 0.05; 1.35 ± 0.06-fold) compared to sham Mg; however, no significant changes were observed for other pro-inflammatory mediators (Figure 1E). By comparison, in the HP, SF significantly activated Mg (*il1β*, 3.18 ± 0.21-fold; *il6*, 3.47 ± 0.35-fold; *tnfα*, 1.92 ± 0.14 fold; and *ccl2*, 2.40 ± 0.27 fold; n = 6, * *p* < 0.05, Figure 1F). In the PFc, we did not observe significant differences between SF and sham Mg in the levels of these mediators (Figure 1G). These findings strongly suggest that three weeks of SF directly activates in vivo microglia in a region-specific manner. We also investigated the levels of pro-inflammatory mediators in the homogenates of the striatum, HP, and PFc of SF and sham mice. Overall, we did not find significant changes in neuroinflammation levels across brain regions with three exceptions: *tnfα* (1.24 ± 0.11-fold) and *ccl2* (1.17 ± 0.08-fold) in the striatum and *il6* (1.29 ± 0.05 fold) in the HP (n = 6, * *p* < 0.05, Figure 1H–J). These results suggested that other types of brain cells could mask the activation status of Mg in the context of relatively short-term SF. We also checked peripheral inflammation levels in the liver, spleen, and gut. SF significantly elevated the expression of all four examined pro-inflammatory mediators in the liver and spleen; however, only *ccl2* was significantly upregulated in the gut (n = 6, * *p* < 0.05, Appendix A). In summary, our results show that three weeks of SF activates microglia in a brain region-specific manner and significantly elevates peripheral inflammation levels.

### 3.2. The Effects of Three Weeks of SF on Microglia, Astrocytes, and CRH Signaling in the HP

Astrocytes have been implicated as a contributor to sleep deprivation mediated neuroinflammation [44,45]. Here, we investigated whether three-weeks of SF could also activate astrocytes. We monitored GFAP and CD11b in the striatum, HP, and PFc of SF and sham brains. SF significantly increased CD11b levels but not GFAP in the HP (n = 6, * *p* < 0.05, Figure 2A). Immunofluorescent results also showed increased Iba1 but not GFAP intensity in the HP of SF mice (n = 6, * *p* < 0.05, Figure 2B). These results indicate that three-weeks of SF has specific effects on Mg but not on astrocytes in the HP. In the striatum and PFc, we did not observe significant changes in CD11b or GFAP levels (n = 6, Figure 2C,D) and signal intensity of Iba1 and GFAP (n = 6, Appendix A). Taken together, three-weeks of SF activates Mg only in the HP and had no effects on astrocytes, suggesting that hippocampal Mg activation is an early event induced by SF.

We next identified mechanisms responsible for SF-mediated Mg activation in the HP. The CRH system is sensitive to CSDs and CRH has been implicated in activating microglia in vitro [38,39]. Therefore, we examined the levels of CRH and its cognate receptors CRHR1 and CRHR2 in the brains of SF and sham mice. CRH, CRHR1, and CRHR2 levels were significantly increased in the HP (n = 6, * *p* < 0.05, Figure 3A) but not in the striatum (Figure 3B). Interestingly, it appeared that the CRH system was downregulated in the PFc of SF mice with significantly lower expression of CRHR1 (n = 6, * *p* < 0.05, Figure 3C). Significant upregulation of CRH mRNA levels was found in the HP of SF mice indicating that upregulation of CRH is probably due to enhanced transcription levels (n = 6, * *p* < 0.05, Appendix A). The concurrent upregulation of the CRH system and Mg activation in the HP of SF mice suggests that the CRH system is involved in SF-mediated Mg activation in vivo.

### 3.3. CRH Activates Mg and Upregulates Gal3 Levels In Vitro

Previous studies indicated that CRH could activate BV2 microglia in vitro [46]. In this study, we employed both BV2 cells and PM to identify signaling underlying CRH-mediated Mg activation. CRH increased il1β and tnfα levels in both BV2 cells and PM (n = 3, * *p* < 0.05, Appendix A) consistent with previous results. CRH significantly increased CD11b levels in both BV2 cells and PM (Appendix A). At the protein level, CRH increased the levels of mIL1β in both BV2 cells and PM (* *p* < 0.05, Appendix A). All these results demonstrated that CRH has the ability to activate Mg in vitro.

Gal3 is highly induced in activated microglia and serves as a neuroinflammation marker in multiple neurodegenerative diseases [47,48]. Since CRH can activate Mg, we explored whether Gal3 is involved in CRH-mediated Mg activation. BV2 cells and PM were cultured and exposed to CRH at varying doses (25–200 nM) for 6 or 24 h followed with protein extraction. In BV2 cells, CRH significantly increased Gal3 levels at all tested dosages at both 6 and 24 h after treatment (* *p* < 0.5, Figure 4A). CRH exerted similar upregulation effects on Gal3 in PM at both time points after treatment (* *p* < 0.05, Figure 4B). These results demonstrated that CRH could upregulate Gal3 levels implying that Gal3 is involved in CRH-mediated Mg. Increased Gal3 could have two different functions based on their different destiny: (1) Gal3 could be secreted and bind to Toll-like receptor (TLR) to activate neighboring Mg; (2) Gal3 could stay inside the cells and form dot-like puncta as an early marker for damaged lysosomes (disrupted lysosomal membrane integrity) [49]. We performed Gal3 immunostaining to explore whether CRH promotes Gal3 dot-like puncta in Mg. The results showed that CRH significantly increased the puncta number of Gal3 after 24 h treatment in all three tested doses in PM (* *p* < 0.05, Figure 4C). CRH induced Gal3 puncta formation as early as 2 h post-treatment and this effect persisted 24 h (* *p* < 0.05, Figure 4D). These results suggest that CRH could induce lysosomal damage through increased lysosomal membrane permeability (LMP).

### 3.4. CRH Dysregulates Autophagy Processes and Induces Lysosome Damage In Vitro

Normal lysosomal function is critical for the completion of autophagy processes and lysosomal damage may lead to the accumulation of autophagosomes. CRH can induce autophagy in neurons which is involved in synapse loss in vitro [50]. Since CRH can induce lysosomal damage in Mg, we explored the effects of CRH on autophagy processes by examining the levels of the autophagosome marker, LC3BII. CRH increased LC3BII levels at both 6- and 24-h after treatment in BV2 cells (* *p* < 0.05, Figure 5A). In PM, CRH similarly upregulated the levels of LC3BII at all tested doses (* *p* < 0.05, Figure 5B). LC3B immunostaining results showed that CRH significantly increased autophagosome formation in PM (* *p* < 0.05, Figure 5C). Additionally, CRH significantly increased the levels of beclin1 and SQSTM1 in PM after 24 h of treatment (* *p* < 0.05, Appendix A). These results showed that that CRH could increase autophagy induction. Cathepsins (Cat) are primarily located within lysosomes and their subcellular location and levels are crucial for lysosomal degradation. Cat can be released from lysosomes into cytoplasm due to increased LMP. Therefore, we monitored Cat B expression in PM with CRH exposure. Immunostaining results showed that at basal conditions, Cat B restricts its location in lysosomes represented by “dot-like” expression. With CRH stimulation, Cat B showed more diffuse expression indicating the translocation from lysosomes into cytoplasm (* *p* < 0.05, Figure 5D). These results demonstrated that CRH can induce lysosomal damages in Mg consistent with the upregulation of autophagosome formation. Additionally, CRH upregulated the expression of Cat B at 6 and 24 h post-treatment (* *p* < 0.05, Figure 5E). We next investigated the effects of CRH on lysosomal biogenesis through examining the levels of transcription factor EB (TFEB) and lysosomal membrane associated protein (LAMP) 1 and 2. There was no obvious difference in the expression of these three molecules after 6 h and 24 h of treatment indicating that CRH has marginal effects on lysosomal biogenesis (Appendix A). Taken together, CRH could induce lysosomal damage and promote autophagy dysregulation.

### 3.5. CRHR2 Blockage Mitigates CRH-Mediated Gal3 Upregulation and Mg Activation In Vitro

CRH exerts its biological effects through binding to CRHR1/2. CRHR2 is highly expressed in both BV2 cells and PM but CRHR1 was barely detected in vitro (data not shown here). CRH has marginal effects on CRHR2 expression in Mg (Figure 6A). We employed the CRHR2 specific antagonist As-2B to assess the biological effects of CRH on microglia. Mg were pre-treated with As-2B (100 nM) for 1 h followed with CRH exposure for another 24 h. Our results showed that CRH increased Gal3 and such regulation was mitigated by As-2B pre-treatment (Figure 6B, * *p* < 0.05 vs. control, # *p* < 0.05, vs. CRH treatment). As-2B could also partially inhibit CRH-mediated upregulation effects on CD11b (Figure 6C, * *p* < 0.05 vs. control, # *p* < 0.05, vs. CRH treatment). In addition, As-2B mitigated the upregulation of mCat B induced by CRH (Figure 6D, * *p* < 0.05 vs. control, # *p* < 0.05, vs. CRH treatment). Overall, As-2B could mitigate CRH-mediated Gal3 upregulation and Mg activation suggesting that CRHR2 plays critical roles in CRH-mediated effects on Mg.

### 3.6. SF Mice Show Increased Gal3 Expression and Autophagy Dysregulation in the Brain HP

We have shown that CRH could increase Gal3 levels and dysregulate autophagy in microglia in vitro. We next explored whether such upregulation also exists in the brains of SF mice. Our results clearly showed that Gal3 increased its expression levels in the HP of SF mice compared to controls but not the striatum or PFc (n = 6, * *p* < 0.05, Figure 7A). Similar upregulation patterns were also found for LC3BII and mCat B in the brains of SF mice (n = 6, * *p* < 0.05, Figure 7B,C). There was increased co-localization of Gal3 and Iba1 in the HP of SF mice (n = 6, Figure 7D). We did not find significant changes in the levels of LAMP1/2 and TFEB in the brains of SF and Sham mice which is consistent with in vitro findings (n = 6, Appendix A). Together, these findings suggest that that Gal3 and autophagy dysregulation are involved in the effect of three weeks of SF on Mg in the HP.

## 4. Discussion

CSDs are prevalent around the world, contributing to the increased risks of multiple neurodegenerative and neuropsychiatric diseases in modern society. Microglial activation has been suggested as a critical factor involved in CSD-mediated brain dysfunction but the underlying mechanisms remain unknown. In this study, we determined that the hippocampal CRH system is sensitive to sleep disturbances and is responsible for CSD-mediated microglial activation in vivo. Mechanistically, CRH upregulates Gal3 expression and induces lysosomal impairment in vitro and in vivo. Overall, our findings suggest a novel mechanism responsible for CSD-mediated microglial activation.

One of our interesting findings is that three-weeks of SF had direct activation effects on purified Mg isolated from the HP but only had marginal effects on neuroinflammation levels of HP homogenates that included all types of brain cells including astrocytes, neurons, oligodendrocytes, endothelial cells, etc. It is possible that other types of brain cells are not sensitive to three weeks of SF and might actually mask the activation effects of SF on Mg (Mg account only 10–15% of all brain cells) when we performed homogenate analysis. Indeed, we did not find astrocytes activation in the three brain regions we examined in SF mice. This discrepancy with previous investigations [43] is probably due to the time period of SF (three weeks of SF vs. two month of SF) or the type of sleep disturbance (SF vs. sleep deprivation) [19]. Regardless, our findings suggest that hippocampal Mg activation is an early event induced by SF and might contribute to CSD-mediated neurological symptoms such as memory and cognitive performance if sleep problems continue.

The CRH system is very sensitive to various types of stress including sleep disturbances. Increased CRH levels/activity has been found in both rodent models of sleep disorders and in patients with sleep problems [51,52]. Classically, CSDs increase the activation of the hypothalamic-pituitary-adrenal (HPA) axis by enhancing CRH levels in the paraventricular nucleus of the hypothalamus. Then, CRH is secreted from CRH producing neurons and acts on the anterior pituitary gland to initiate a cascade of stress responses [53,54]. Recently, accumulating evidence indicates that the CRH system (CRH and CRHRs) is also highly expressed in extra HPA regions including the STR, HP, PFc and is sensitive to sleep problems [31,32,33]. In addition, amygdala CRH has been well-investigated with respect to the effects on fear-induced reductions in sleep indicating reciprocal interactions between the CRH system and sleep disturbances [55,56,57]. In this study, we demonstrated that CRH system activity was increased in the HP by three weeks of SF, adding more evidence that CRH in extra HPA regions is also tightly regulated by sleep disturbances. Interestingly, we did not observe CRH system dysregulation in the STR and PFc indicating that the hippocampal CRH system may be more sensitive than other brain regions and that the alterations in the hippocampal CRH system is an early event induced by sleep disturbances.

The mechanisms by which CSDs induce microglial activation and neuroinflammation in the brain are unclear. Endogenous molecules that could be tightly regulated by CSDs and also function as upstream signals for microglial activation have not been identified. Previous studies showed that CRHR 1 and 2 are expressed in microglial cells with relatively higher levels of CRHR2 [58] and that CRH was capable of increasing mRNA levels of proinflammatory mediators in microglia in vitro [46]. Our results further demonstrate the ability of CRH to activate microglia by showing that CRH upregulates CD11b levels and mILβ in both BV2 cells and PM. Of note, we observed high correlation between CRH upregulation and microglial activation in the HP of SF mice. Therefore, our results strongly suggest that the CRH system could serve as a link bridging CSDs and neuroinflammation in vivo.

In searching for down-stream effectors of CRH, we have been focusing on Gal3 which is highly induced in activated microglia and plays critical roles in multiple neuroinflammation and neurovegetative diseases [47,48]. Gal3 belongs to the galectin family which includes 15 members (Gal1-15). Among those members, Gal3 is unique in its “chimera type” structure constituting of a single C-terminal carbohydrate recognition domain and a non-lectin collagen-like N-terminal region [59]. Gal3 could be released from activated microglia and functions as an endogenous ligand to bind Toll-like receptor 4 in neighboring microglia to amply immune responses [60,61]. Gal3 has also been identified as a Trem2 ligand involved in the regulation of the inflammatory response in Alzheimer’s disease [62]. Gal3 has been suggested to be a novel neuroinflammation marker [63] and increased Gal3 levels have been identified in multiple neurodegenerative diseases including Parkinson’s disease, Alzheimer’s disease, Huntington’s disease, and amyotrophic lateral sclerosis [47,48]. In vitro, Gal3 can be induced by several stimulators including LPS [64] and amyloid [65]. The roles of Gal3 in neuroinflammation in the context of sleep disorders are not known. In this study, we showed, in vitro, that CRH could upregulate Gal3 expression in BV2 and PM and that CRHR2 blockage could reverse CRH-mediated Gal3 upregulation. Additionally, CRH could increase Gal3 puncta formation in vitro. These results demonstrate that Gal3 is a novel substrate regulated by CRH and suggest that Gal3 may be involved in CRH-mediated microglial activation. Gal3 upregulation was also identified in the brains of SF mice indicating that Gal3 might play critical roles in CSD-mediated neuroinflammation in vivo. The mechanisms responsible for Gal3 upregulation by external stimuli are mostly unknown though previous investigations showed that miR-124 could target Gal3 and inhibit Gal3 expression [66]. Since miR-124 is abundantly expressed in resting microglia and decreased in activated microglia [67,68], it could be worthwhile to explore whether CRH could decrease miR-124 levels leading to Gal3 upregulation and we have an ongoing project for this purpose.

CRH is capable of interacting with autophagy processes [69,70,71] and autophagy dysregulation has demonstrated roles in regulating inflammation and microglial activation [72,73]. We showed that CRH upregulated the levels of becin1, LC3BII, and p62 in PM as well as the increased formation of the autophagosome indicating that CRH could dysregulate autophagy by blocking autophagosome and lysosomal fusion. Normal lysosome degradation is critical for the completion of the autophagy process. In addition to being a neuroinflammation marker, Gal3 is also a well-accepted earlier marker for lysosomal damages [49]. When a lysosome is damaged or shows increased membrane permeability, Gal3 could gather around the disrupted lysosomes forming dot-like pattern. Such a response could initiate lysophagy to remove the damaged lysosomes. Interestingly, our results showed that CRH could induce a Gal3 dot-like expression pattern in microglial cells as early as 2 h post-treatment and persist 24 h post-treatment. In parallel, lysosomal Cat B showed a more diffuse expression pattern in cells implying the translocation from lysosomes into cytoplasm. Taken together, these results indicate that CRH could impair lysosome degradation by inducing LMP which ultimately results in autophagy dysregulation. Therefore, autophagy dysregulation is also involved in CRH-mediated Mg activation.

We showed that CRH could upregulate Gal3 levels and lysosomal damage in Mg. However, the sequence for these two events is not clear. They could happen simultaneously or consecutively. There is no evidence that Gal3 can directly induce lysosomal damage though it is an earlier maker for such damage. It is possible that Gal3 activates the TLR4/NF-κB pathway or other unknown signals leading to increased levels of ROS which is capable of inducing LMP. Another possibility is that lysosomal damage come first which induces Gal3 expression. In each of these possibilities, Gal3 and lysosomal damage might work synergistically leading to Mg activation.

## 5. Conclusions

Three-weeks of SF induced hippocampal Mg activation, and increased CRH system activation underlies Mg activation in vitro and in vivo. Mechanistically, CRH could upregulate Gal3 and induce autophagy dysregulation through lysosomal dysfunction. Thus, targeting the CRH system might provide a promising treatment avenue for ameliorating CSD-mediated neuroinflammation and related neurological symptoms.

## Figures and Tables

**Figure 1 cells-12-00160-f001:**
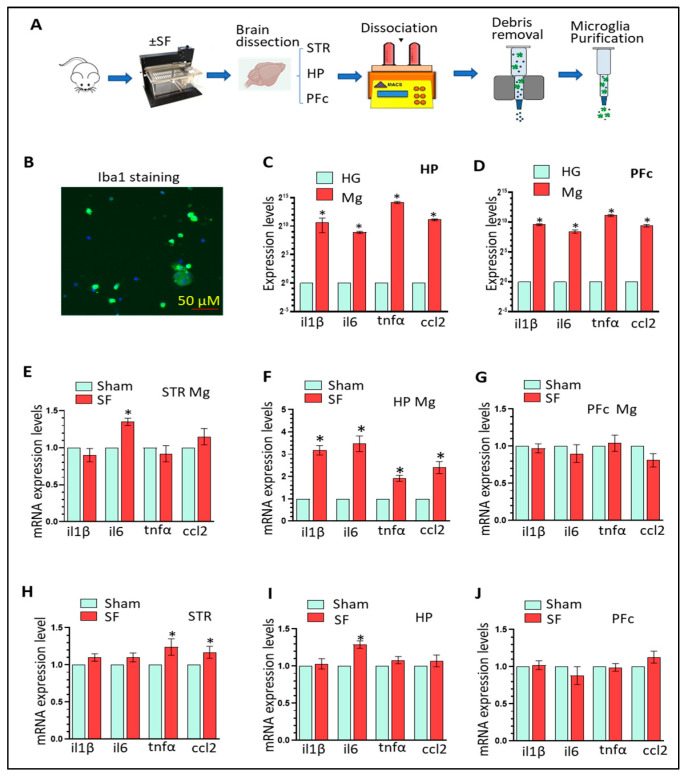
Three-weeks of SF activates microglia in the HP. (**A**): Schematic for the experimental design. (**B**): Iba1 immunostaining for purified adult Mg. (**C**,**D**): Significant enrichment of pro-inflammatory mediators in purified Mg compared to homogenates from the HP and PFc (n = 6, * *p* < 0.05). (**E**–**G**): levels of four pro-inflammatory mediators in purified Mg isolated from the striatum, HP, and PFc of SF and sham mice. (n = 6, * *p* < 0.05, sham vs. SF). (**H**–**J**): levels of four pro-inflammatory mediators in the homogenates prepared from the striatum, HP, and PFc of SF and sham mice (n = 6, * *p* < 0.05, sham vs. SF).

**Figure 2 cells-12-00160-f002:**
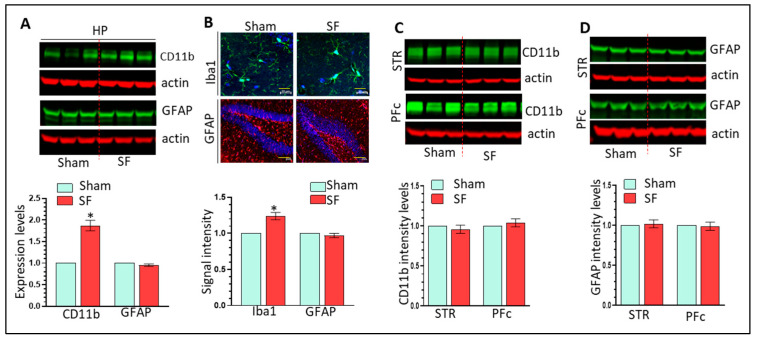
Three-weeks of SF activates microglia but not astrocytes in the brain. (**A**): three-weeks of SF increases CD11 levels but not GFAP levels in the HP (n = 6, * *p* < 0.05); (**B**): three-weeks of SF increases Iba1 signal intensity but not GFAP levels in the HP (n = 6, * *p* < 0.05, scale bar = 20 µ); (**C**,**D**): three-weeks of SF does not increase CD11 and GFAP levels in the striatum or PFc (n = 6).

**Figure 3 cells-12-00160-f003:**
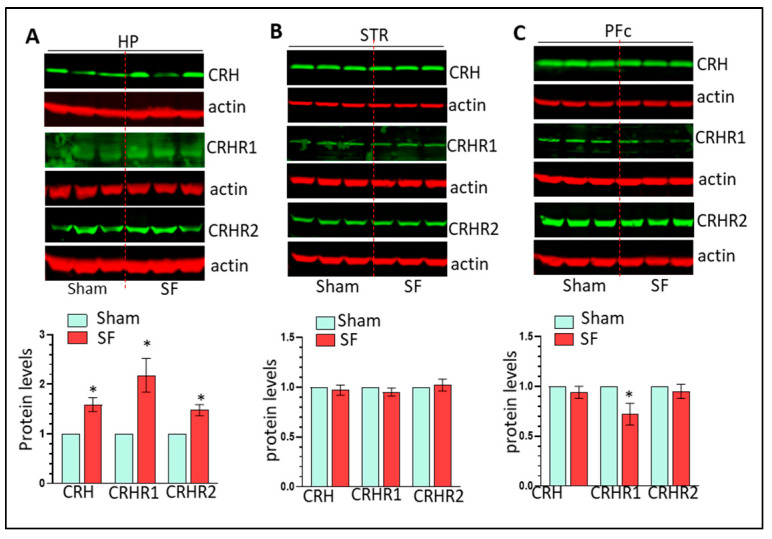
Three-weeks of SF selectively increases CRH system activity in the brain. (**A**): three-weeks of SF increases the levels of CRH, CRHR1, and CRHR2 levels in the HP (n = 6, * *p* < 0.05); (**B**,**C**): three-weeks of SF has no effects on CRH system in the striatum and PFc.

**Figure 4 cells-12-00160-f004:**
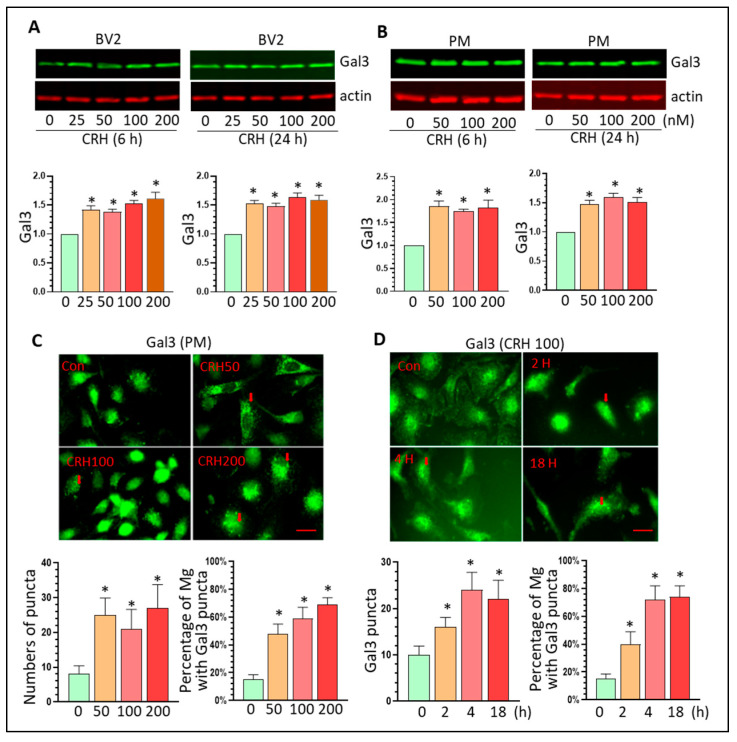
CRH increases Gal3 expression in vitro. (**A**): CRH increases Gal3 levels in BV2 cells (* *p* < 0.05); (**B**): CRH increases Gal3 levels in PM (* *p* < 0.05); (**C**,**D**): CRH increases the numbers of Gal3 puncta in Mg and increase the percentage of Gal3 puncta positive in Mg (* *p* < 0.05, scale bar = 10 µ). For Western blots, all experiments were independently repeated at least three times for statistical analysis.

**Figure 5 cells-12-00160-f005:**
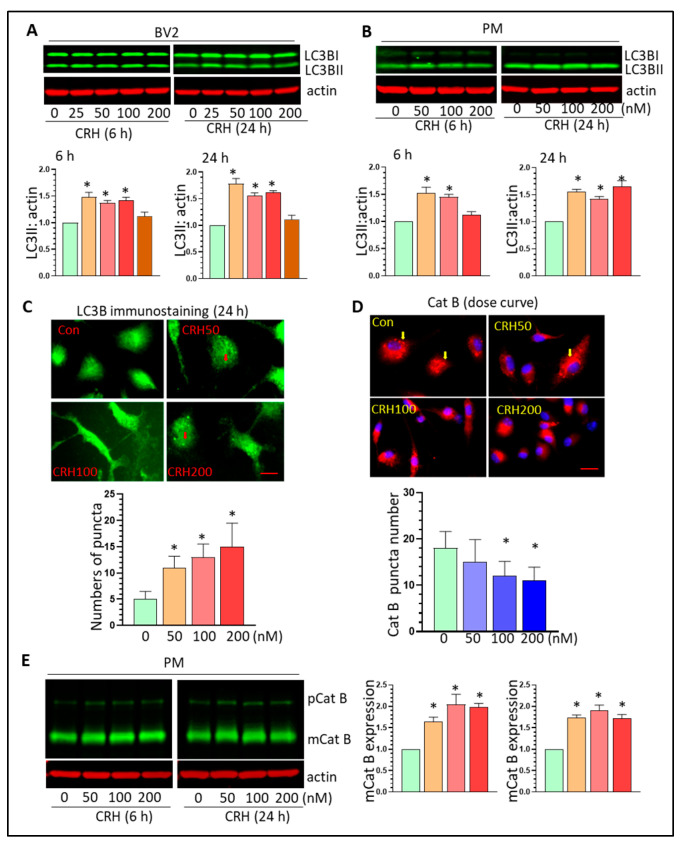
CRH increases autophagosome formation and induces Cat B leakage. (**A**,**B**): CRH increases LC3BII levels in BV2 cells and PM (* *p* < 0.05); (**C**): CRH increases autophagosome formation in PM (* *p* < 0.05, scale bar = 10 µ); (**D**): CRH increases Cat B release in PM (* *p* < 0.05, scale bar = 10 µ); (**E**): CRH increases mCat B levels in PM (* *p* < 0.05). All experiments were independently repeated at least three times for statistical analysis.

**Figure 6 cells-12-00160-f006:**
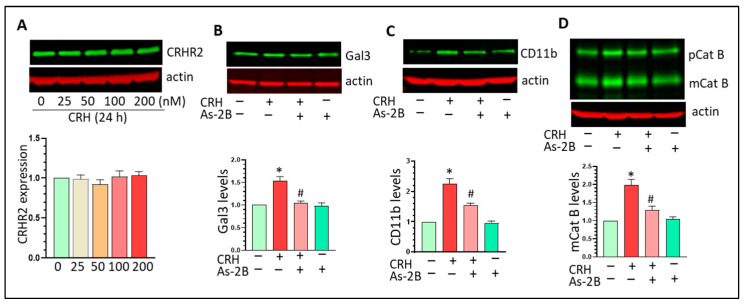
CRHR2 blockage mitigates CRH-mediated effects on Mg. (**A**): CRH has no effects on CRHR2 levels in BV2 cells; (**B**): As-2B mitigates CRH-mediated upregulation ofGal3 (* *p* < 0.05 vs. control, # *p* < 0.05, vs. CRH treatment); (**C**): As-2B mitigates CRH-mediated upregulation effects on CD11n (* *p* < 0.05 vs. control, # *p* < 0.05, vs. CRH treatment); (**D**): As-2B mitigates CRH-mediated upregulation effects on mCat B (* *p* < 0.05 vs. control, # *p* < 0.05, vs. CRH treatment). All experiments were independently repeated at least three times for statistical analysis.

**Figure 7 cells-12-00160-f007:**
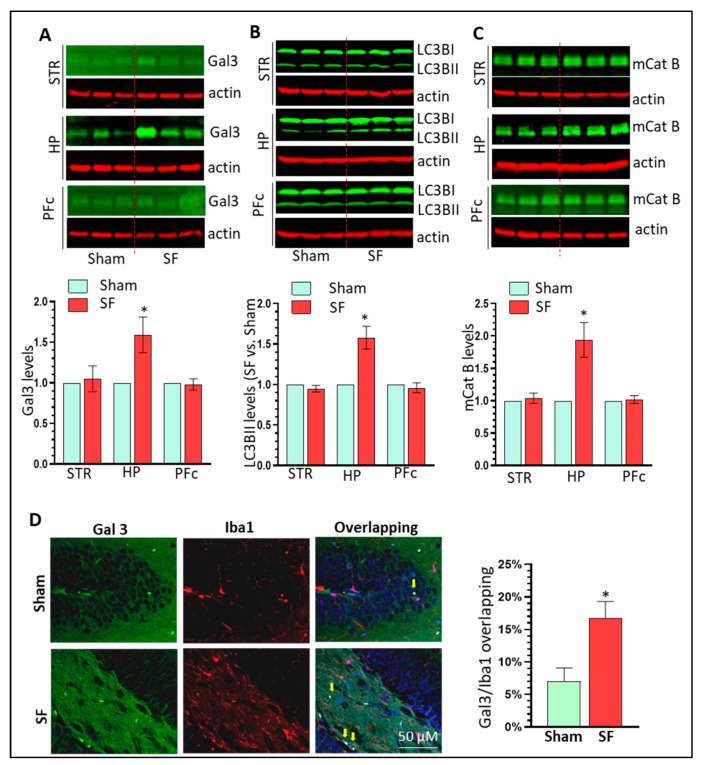
SF mice show increased Gal3 expression and autophagy dysregulation in the brain HP. (**A**): Three weeks of SF increased Gal3 levels in the HP but not in the striatum and PFc (n = 6, * *p* < 0.05); (**B**): Three weeks of SF increased LC3BII levels in the HP but not in the striatum and PFc (n = 6, * *p* < 0.05); (**C**): Three weeks of SF increased mCat B levels in the HP but not in the striatum and PFc (n = 6, * *p* < 0.05); (**D**): Increased co-localization of Gal3 and Iba1 in the HP of SF mice compared to control mice (scale bar = 50 µ).

## Data Availability

Not applicable.

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
