# Peer review of "Sleep-Disturbance-Induced Microglial Activation Involves CRH-Mediated Galectin 3 and Autophagy Dysregulation"

_cells, 2022, doi:10.3390/cells12010160_

Round 1
Reviewer 1 Report
The authors reported in the article the results of an interesting experiment, aimed at clarifying the influence of sleep disruption on brain Microglia and Astrocytes. They well explained the general interest of this research in the introduction, highlighting the prevalence of the sleep disorders as well as the pivotal role of Microglia in inflammatory processes in the brain. To do this, they conducted an experiment with animals of a well-known mice strain by comparing the microglia parameters of sleep-disturbed mice with controls. The indicators they chose are appropriate and reliable for describing microglia reactions and for interpreting the biological meaning of these changes. All results without exception, converge towards a harmful action of the type of Microglia activation induced by sleep disruption, also highlighting some possible biochemical mechanisms. The experiments were well planned and executed, with a wealth of data beautifully displayed in the results section. The discussion examined the results and provides us with further interpretative hypotheses that can be deduced from the data. Indeed the discussion followed the subsequent sections of the results: the microglia activation effects, CRH, Gal3, autophagy. It is true that at least part of such noticeable results may be due to the strong method adopted for sleep disruption, while a more gentle way of sleep disruption could have given different results ( such as those described in : Yan Zhu, Guanxia Zhan, Polina Fenik, Madison Brandes, Patrick Bell, Noelle Francois, Katherine Shulman, Sigrid Veasey Chronic Sleep Disruption Advances the Temporal Progression of Tauopathy in P301S Mutant Mice Journal of Neuroscience 15 October 2018, 0275-18; DOI: 10.1523/JNEUROSCI.0275-18.2018 ).
Conclusions are well supported by the results.
At the end an interesting research with new data and new interesting hypotheses.
Just one remark: the sample sizes overall and in each experiment and analysis are not clearly reported, This is one of the main point of the “ARRIVE guidelines 2.0” : Sample size
2a. Specify the exact number of experimental units allocated to each group, and the total number in each experiment. Also indicate the total number of animals used.
c. For each analysis, report the exact value of n in each experimental group.
Author Response
Reviewer 1:
The authors reported in the article the results of an interesting experiment, aimed at clarifying the influence of sleep disruption on brain Microglia and Astrocytes. They well explained the general interest of this research in the introduction, highlighting the prevalence of the sleep disorders as well as the pivotal role of Microglia in inflammatory processes in the brain. To do this, they conducted an experiment with animals of a well-known mice strain by comparing the microglia parameters of sleep-disturbed mice with controls. The indicators they chose are appropriate and reliable for describing microglia reactions and for interpreting the biological meaning of these changes. All results without exception, converge towards a harmful action of the type of Microglia activation induced by sleep disruption, also highlighting some possible biochemical mechanisms. The experiments were well planned and executed, with a wealth of data beautifully displayed in the results section. The discussion examined the results and provides us with further interpretative hypotheses that can be deduced from the data. Indeed the discussion followed the subsequent sections of the results: the microglia activation effects, CRH, Gal3, autophagy. It is true that at least part of such noticeable results may be due to the strong method adopted for sleep disruption, while a more gentle way of sleep disruption could have given different results ( such as those described in : Yan Zhu, Guanxia Zhan, Polina Fenik, Madison Brandes, Patrick Bell, Noelle Francois, Katherine Shulman, Sigrid Veasey Chronic Sleep Disruption Advances the Temporal Progression of Tauopathy in P301S Mutant Mice Journal of Neuroscience 15 October 2018, 0275-18; DOI: 10.1523 /JNEUROSCI. 0275 -18. 2018).
Conclusions are well supported by the results.
At the end an interesting research with new data and new interesting hypotheses.
Response: The authors would like to thank the reviewer for his/her time and effort in reviewing our manuscript as well as the positive comments. Below are the point-to-point responses.
Just one remark: the sample sizes overall and in each experiment and analysis are not clearly reported, This is one of the main point of the “ARRIVE guidelines 2.0”: Sample size 2a. Specify the exact number of experimental units allocated to each group, and the total number in each experiment. Also indicate the total number of animals used. c. For each analysis, report the exact value of n in each experimental group.
Response: We thank the reviewer for this comment. We added sample size (the exact of n) in each experimental group.
Reviewer 2 Report
The submission from Liyang Guo He et al. reports an interesting work that three-weeks of sleep fragmentation (SF) induces microglial activation in mouse hippocampus through CRH system-mediated galectin 3 and autophagy dysregulation. However, there are some gaps between CRH/ CRHR2 and upregulation of Gal3, and Gal3 increase and autophagy dysregulation.
1. In Figure 2B, method for analysis of Iba1 signal intensity is needed and the name of Y-axis is not right.
2. The quality of the CRHR1 western blot in Figure 3A should be improved.
3. In Figure 4C and D, the method of counting the numbers of Gal3 puncta is needed, and the percentage of microglia with Gal3 puncta is also an important support.
4. In Figure 7D, it is better to show the percentage of Gal3 and Iba1 co-localized microglia.
Author Response
Reviewer 2:
The submission from Liyang Guo et al. reports an interesting work that three-weeks of sleep fragmentation (SF) induces microglial activation in mouse hippocampus through CRH system-mediated galectin 3 and autophagy dysregulation. However, there are some gaps between CRH/CRHR2 and upregulation of Gal3, and Gal3 increase and autophagy dysregulation.
- In Figure 2B, method for analysis of Iba1 signal intensity is needed and the name of Y-axis is not right.
Response: Yes. We added the method for analysis of Iba1 signal intensity and corrected the Y-axis.
- The quality of the CRHR1 western blot in Figure 3A should be improved.
Response: We repeated the western blots for CRHR1 and replaced it with a new one.
- In Figure 4C and D, the method of counting the numbers of Gal3 puncta is needed, and the percentage of microglia with Gal3 puncta is also an important support.
Response: Yes, we added the method of counting the Gal3 puncta and calculated the percentage of microglia with Gal3.
- In Figure 7D, it is better to show the percentage of Gal3 and Iba1 co-localized microglia.
Response: Yes. We added the percentage of Gal3 and Iba1 colocalization in microglia.
Round 2
Reviewer 2 Report
The manuscript has been much improved. Regarding the concern (the gap between CRH/CRHR2 and Gal3 upregulation), although the authors hypothesize that miR-124 may be the mediator, are other molecules involved? Or do you have any experimental support?
Author Response
Reviewer 2:
The manuscript has been much improved. Regarding the concern (the gap between CRH/CRHR2 and Gal3 upregulation), although the authors hypothesize that miR-124 may be the mediator, are other molecules involved? Or do you have any experimental support?
Response: We thank the reviewer for the positive comment. We recognize the gap between CRH/CRHR2 and Gal3 upregulation and are trying to identify molecules which could bridge it. We have an ongoing project exploring the involvement of miR-124 in CRH-mediated Gal3 upregulation and our preliminary data shows that CRH could decrease miR-124 levels. However, at this point, we prefer to only speculate until our data provide a clearer answer. Hopefully, we can fully address this question in a future contribution.